IPPP/21/30, LU-TP 21-41, MCnet-21-16

# Coloring mixed QCD/QED evolution

Leif Gellersen,[1] Stefan Prestel,[1] and Michael Spannowsky[2]

[1]*Department of Astronomy and Theoretical Physics,*
*Lund University, S-223 62 Lund, Sweden*
[2]*Institute for Particle Physics Phenomenology, Department of Physics,*
*Durham University, DH1 3LE, United Kingdom*

Parton showers are crucial components of high-energy physics calculations. Improving their modelling of QCD is an active research area since shower approximations are stumbling blocks for precision event generators. Naively, the interference between sub-dominant Standard-Model interactions and QCD can be of similar size to subleading QCD corrections. This article assesses the impact of QCD/QED interference effects in parton showers, by developing a sophisticated shower including QED, QCD at fixed color, and employing complete tree-level matrix element corrections for individual $N_C = 3$ color configurations to embed interference. The resulting simulation indicates that QCD/QED interference effects are small for a simple test case and dwarfed by electro-weak resonance effects.

## I.  INTRODUCTION

General-Purpose Monte-Carlo Event Generators (MCEGs) are important tools for collider physics, especially to exploit the potential of current and future measurements at the Large Hadron Collider [1]. In particular, a program of indirect searches for phenomena outside the well-established Standard Model will require comprehensive high-precision simulations. Typically, parton showers – the effect of dressing fast-moving charges with soft and/or collinear radiation – are a non-negligible component of the physics modelling of MCEGs; and their uncertainty budget. An improved parton shower, and thus improved error budget will allow for more aggressive strategies for searching for new physics through indirect limit setting. Because of this, many improvements to QCD parton showers have been proposed. In particular, higher-order showers [2–6], showers at higher logarithmic accuracy [7–11] and color-correct QCD showers (that retain full color correlators when calculating emission rates) [12–19] are active research areas. Beyond QCD, electroweak showers have been discussed [20–23] and QED effects are typically included.

Once the complete gauge and matter content of the Standard Model becomes part of the shower evolution, assessing the numerical impact of various approximations becomes complicated. In particular, interference between intermediate states with different gauge bosons (and hence gauge couplings) can have a similar or even more significant impact than corrections to the strong-coupling contributions. For example, it is not apparent that the effect of QCD/QED mixing is subdominant to subleading-color QCD corrections. We want to address this question in this article.

To understand the reasoning, it is useful to consider the simple partonic scattering $e^+e^- \to q\bar{q}q\bar{q}$. For this process, both diagrams with intermediate gluons and with intermediate photons contribute. Two distinct ways of assigning and connecting colors are allowed within the color flow basis, so that the squared matrix element may be sketched as

$$|\mathcal{M}|^2 \propto \sum_{\text{colors}} \left| \dots \right|^2 \tag{1}$$

$$= 8 \left| \dots \right|^2 + 8 \left| \dots \right|^2 - \frac{8}{3} \cdot 2\,\mathrm{Re}\left\{ \dots \right\} + 8 \cdot 2\,\mathrm{Re}\left\{ \dots \right\} + 8 \cdot 2\,\mathrm{Re}\left\{ \dots \right\} + \mathcal{O}\left(\alpha_{\text{em}}^2\right) \ ,$$

where the flavor and momentum assignment for all diagrams is shown in the first diagram. Due to the interference between different gluonic diagrams, the subleading color contribution is proportional to $-\frac{8}{3}\alpha_s\alpha_s$. Ignoring electric charge factors of $\mathcal{O}(1)$, the contributions from the interference between QED and QCD are $\propto 16\,\alpha_{\text{em}}\alpha_s = 16\frac{\alpha_{\text{em}}}{\alpha_s}\alpha_s\alpha_s$. Thus, for typical values of $\alpha_s$ and $\alpha_{\text{em}}$, subleading-color and QCD/QED interference contributions can be of similar magnitude, and even accidental cancellations might occur.

This simple example motivates the more careful assessment of QCD/QED interference effects in this article. This is not necessarily straightforward: interference effects can be sensitive to specific color configurations, to the extent that an approximation at the color-summed level alone might obscure interesting effects. Thus, a QCD parton shower that can sample fixed ($N_c = 3$) color structures may be required. Consequently, we present the first fixed-color

parton shower implementation within PYTHIA [24], corrected, to our knowledge for the first time, by color-specific kinematic matrix element correction factors at $N_c = 3$. Our approach forms a solid baseline for an interleaved QCD/QED parton shower evolution that incorporates interference effects through iterated matrix element corrections. In passing, it is worth mentioning that the resulting implementation should, in the future, enable event generator improvements through matching or merging methods without resorting to the $N_C \to \infty$ limit at any intermediate stage. Furthermore, the sophisticated implementation allows an assessment of the size of QCD/QED interference effects relative to subleading color corrections alone. We hope that this will either question or support efforts focussing on QCD corrections alone.

To remain as demonstrative as possible, we will use $e^+e^- \to$ jets as our test case, and in particular study the relevant effects in isolation, by focussing on the modelling of $e^+e^- \to q\bar{q}q\bar{q}$ and $e^+e^- \to q\bar{q}q'\bar{q}'$ events. At LEP, four-parton states have been used to study the effect of color reconnection, with some emphasis on the effect of non-perturbative modelling on the extraction of the W-boson line shape [25]. The effect of QCD/EW/QED interference has also been studied at hadron colliders long ago [26, 27], mostly in (di)jet production events. These studies were limited to the hardest interaction, modelled with fixed-order perturbation theory. Compared to these earlier works, we focus on all-order perturbative effects here.

## II. IMPLEMENTATION

To study the effects of fixed-color parton shower evolution, QED parton shower emissions and their interplay, we implement a combination of those features based on the DIRE parton shower [28] plugin for PYTHIA [24]. The interference between fixed color QCD and QED evolution enters the Monte Carlo simulation by implementing iterative matrix element corrections for the shower based on [29] and adapted to go beyond leading color configurations.

This section provides a brief overview of the DIRE parton shower and its QED shower implementation. Then, we explain in detail how we incorporate the fixed color shower evolution based on [14] and detail the adaption of iterative matrix element corrections to allow for their use with fixed color evolution. At the end of the section, we summarise the combined transition probability.

### A. Dire and its QED shower

QCD evolution within the DIRE shower is based on the dipole picture of [30, 31], which factorizes a generic real-emission matrix element $\mathcal{M}_n$ in soft or collinear limits as

$$|\mathcal{M}_{n+1}|^2 \simeq - \sum_{ij,k \neq ij} \langle \mathcal{M}_n | \frac{\mathbf{T}_k \mathbf{T}_{ij}}{\mathbf{T}_{ij}^2} \mathcal{V}_{ij,k} | \mathcal{M}_n \rangle \xrightarrow{\text{average over color and spin}} |\mathcal{M}_n|^2 \sum_{ij,k \neq ij} \frac{1}{(p_i + p_j)^2 - m_{ij}^2} 8\pi \alpha_s P_{i,j,k} . \quad (2)$$

Taking the $N_c \to \infty$ limit, and discarding spin correlations between $\mathcal{M}_n$ and $\mathcal{V}$, leads to a complete factorization of the leading-color dipole kernels $P_{i,j,k}$. In this limit, the sets of pairs of radiator $(ij)$ and recoiler $(k)$ are determined by the color connection in the $N_c \to \infty$ limit.

The evolution variable for different color connections is chosen such that an emission of particle $j$ in a splitting $ij, k \to i, j, k$ will occur at the same evolution scale, independent of whether $ij$ or $k$ emitted the particle $j$. However, the complete radiation matrix element is partial-fractioned according to [30, 31], so that the splitting kernels do rely on the identification of $ij$ or $k$ branching. This parton-shower like behavior clearly identifies the collinear sectors in which splitting kernels act, designed to simplify extensions of the shower evolution beyond leading order.

An extension of this framework to incorporate QED transitions was sketched in [32], yet not documented in detail. Thus, we will discuss relevant aspects here. The QED shower in DIRE combines the massive dipole splitting kernels of [31] with the appropriate charge correlators for QED, e.g. given in [33]. The QED charge correlators rely on the flavor of both the radiating particle and the recoiling partner. Each dipole splitting kernel contains a fraction of the (eikonal) soft photon radiation pattern – with the complementary fraction being assigned to emissions off the recoiling particle. Since eikonal radiation patterns in QED arise between any two charges, each possible combination of charged particles should be considered as recoiler-radiator pair, as illustrated in Figure 1. This may lead to negative charge correlators, i.e. a-priori non-suppressed negative-valued splitting kernels. This complication is handled by employing weighted Sudakov veto algorithm techniques [34–36].

The presence of multiple recoilers for radiation from a fixed particle is natural for soft-photon radiation. The treatment of collinear radiation is less obvious. The factorization in Eq. 2 dictates that in the case of QCD at fixed color, the complete collinear radiation pattern should be assigned to each radiator-recoiler pair. The similarity of the set of dipoles between QED and fixed-color QCD suggests a similar strategy for photon radiation – which will

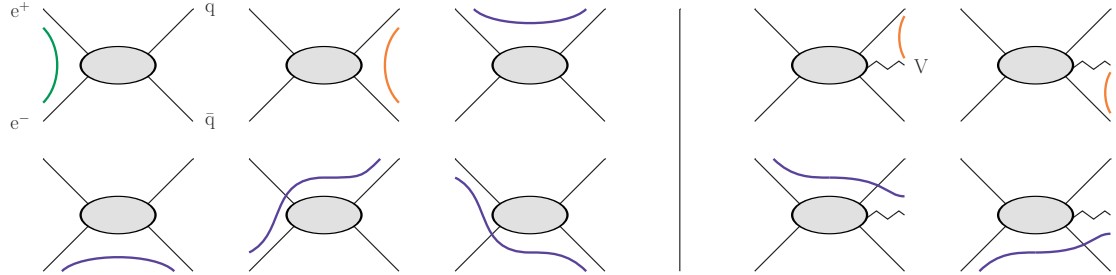

FIG. 1: Dipole configurations for $e^+e^- \to q\bar{q}$ states with and without an additional massless vector boson V (i.e. gluon or photon). Green lines indicate dipoles spanned between initial-state particles, red lines dipoles spanned between final-state particles, and blue lines dipoles spanned between one initial and one final-state particle. In the case of QCD, only a small subset of all possible dipoles – the red ones – is employed. Leading-color QCD parton showers would only employ the two red dipoles in the right panel when showering $e^+e^- \to q\bar{q}g$ states, while subleading-color showers would retain the red dipole spanned between q and $\bar{q}$ in the left panel. QED showers will, in general, employ the complete set of dipoles. In this sense, the set of QED dipoles for $e^+e^- \to q\bar{q}$ states is identical to the set of QCD dipoles obtained for $q\bar{q} \to q'\bar{q}'$ in a fixed-color shower. In the assessment of interferences below, only the configurations from the red dipoles will be used in both the QED and QCD cases, so that a misinterpretation of results based on different phase space coverage of the QCD and QED showers can be avoided.

be employed here. Nevertheless, it is worth remembering that Eq. 2 only aims to approximate the matrix-element in the strict soft or collinear limits, and is ambiguous for non-vanishing transverse momentum. We will employ a slight variation of the functional form of $P_{i,j,k}$ for QED emissions, using

$$P_{i=q,j=\gamma,k} = \frac{-\eta_k \eta_{ij}}{\eta_{ij}^2} \left( \frac{2z(1-z)}{(1-z)^2 + \kappa^2} + (1-z) \right) = \frac{-\eta_k \eta_{ij}}{\eta_{ij}^2} \left( \left[ \frac{2(1-z)}{(1-z)^2 + \kappa^2} - (1+z) \right] + \frac{2\kappa^2}{(1-z)^2 + \kappa^2} \right) , \quad (3)$$

with $z$ and $\kappa$ defined in [28], and where $\eta_a = Q_a \theta_a$ includes the (electric) charge $Q_a$ of particle $a$, and a sign $\theta_a = -1\,(+1)$ if particle $a$ is in the initial (final) state. The first term in parentheses after the second equality is the kinematic part of the conventional form of $P_{i=q,j=g,k}$. This variation still contains appropriate singularities in the soft and collinear limits (defined by $\kappa \to 0$), but will increase the rate of high transverse-momentum photon emissions. The impact of the latter choice will allow assessing the modelling of the full matrix element by the shower below. Note that the functional form of $z$ in terms of the momenta after the splitting depends on the choice of recoiler, which in turn depends on the phase-space mapping used to construct the post-branching momenta. We use the mappings detailed in [28], which distinguish between initial-initial, final-final, initial-final and final-initial cases. Interestingly, for the process illustrated in Figure 1, the initial-final and final-initial phase-space mapping strategies will lead to changes to one of the quark momenta, leading, overall, to a shift in the reconstructed mass of an intermediate state.

Note that the definition and value of $z$ in Eq. 3 will depend on the phase-space mapping strategy – each dipole employs a different definition of "collinear" determined by transverse momenta in the dipole rest frame. These different definitions further motivate assigning each dipole the full collinear radiation pattern. Another conceivable variation of the QED splitting kernel would be to employ

$$\overline{P}_{i=q,j=\gamma,k} = \frac{-\eta_k \eta_{ij}}{\eta_{ij}^2} \frac{2(1-z)}{(1-z)^2 + \kappa^2} - Q_{ij}^2 f_{qq}(\text{recoilers})(1+\bar{z}) , \quad (4)$$

where $\bar{z}$ is independent of the recoiler, e.g. $\bar{z} = \frac{E_j}{E_i + E_j}$, and $f_{qq}(\text{recoilers})$ is a function to ensure that the correct collinear pattern is reproduced by the sum over all possible dipoles. Although interesting, we will not consider this form further, but would like to assert that an inadequate modelling of emissions will be exposed and corrected by matrix element corrections, see section II C.

The choice of recoiler for $\gamma \to f\bar{f}$ splittings is less motivated, given that the splitting does not contain soft singularities. Thus, in principle, any single particle or set of particles could be invoked as recoiler in the generation of post-branching momenta. However, the simultaneous correlated emission of a soft $f\bar{f}$ pair *does* contain soft singularities. Based on this reason, DIRE will consider any electrically charged external particle as viable recoiler. The dipole splitting function for $\gamma \to f\bar{f}$ is then

$$\overline{P}_{i=\gamma,j=f,k} = Q_f^2 f_{\gamma f}(\text{recoilers})(\bar{z}^2 + (1-\bar{z})^2) , \quad (5)$$

where $Q_f$ is the electric charge of the fermion, $\bar{z} = \frac{E_j}{E_i + E_j}$, and $f_{\gamma f}(\text{recoilers}) = 1/[\#\text{recoilers}]$, as suggested in [33].

## B.   Fixed Color shower

The fixed color shower implementation we are using is based on the ideas of stochastically sampling color configurations as described in [14]. The partons in individual events receive fixed color indices $1, 2, 3$ in the fundamental representation for individual events, and the sum over events corresponds to an MC color sum. It goes beyond leading color dipole showers by introducing color-specific splitting kernels that incorporate the effect of different color structures and their interference according to eq. (2).

The color flow basis [37] offers a very convenient way of implementing color sampling, since it allows to decompose a gluon propagator into a nonet and a singlet contribution

$$\langle (\mathcal{A}_\mu)_{j_1}^{i_1} (\mathcal{A}_\nu)_{j_2}^{i_2} \rangle \propto \overbrace{\delta_{j_2}^{i_1} \delta_{j_1}^{i_2}}^{\text{nonet}} - \frac{1}{N_{\text{C}}} \overbrace{\delta_{j_1}^{i_1} \delta_{j_2}^{i_2}}^{\text{singlet}} . \tag{6}$$

Following the notation of [38], we can then represent color-flows by basis tensors

$$|\sigma\rangle = \left| \begin{matrix} 1 & 2 & \ldots & n \\ \sigma(1) & \sigma(2) & \ldots & \sigma(n) \end{matrix} \right\rangle = \delta_{\bar{c}_\sigma(1)}^{c_1} \delta_{\bar{c}_\sigma(2)}^{c_2} \ldots \delta_{\bar{c}_\sigma(n)}^{c_n} , \tag{7}$$

where the permutation $\sigma$ denotes the flow of colors connecting the external legs $1, 2, \ldots, n$, and the fundamental and anti-fundamental color indices $c_i$ and $\bar{c}_{\sigma(i)}$ run from 1 to $N_{\text{C}} = 3$ for external partons carrying (anti-)color, and take the value 0 if not. This allows us to write the color operators

$$\mathbf{T}_i = \boldsymbol{\tau}_i^c + \boldsymbol{\tau}_i^{\bar{c}} + \boldsymbol{\tau}_i^{s_c} \qquad \text{where} \qquad \begin{aligned} \boldsymbol{\tau}_i^c &= \lambda_i \mathbf{t}_{i,c} \\ \boldsymbol{\tau}_i^{\bar{c}} &= -\bar{\lambda}_i \bar{\mathbf{t}}_{i,\bar{c}} \\ \boldsymbol{\tau}_i^{s_c} &= -\frac{1}{N_{\text{C}}} (\lambda_i - \bar{\lambda}_i) \mathbf{s}_c , \end{aligned} \tag{8}$$

and where $\lambda_i$ takes the value $\sqrt{T_{\text{R}}}$ if parton $i$ carries a color and 0 if it does not, and $\bar{\lambda}_i$ takes the value $\sqrt{T_{\text{R}}}$ if parton $i$ carries an anti-color and 0 if not. The operators $\mathbf{t}$, $\bar{\mathbf{t}}$ and $\mathbf{s}$ act on the color basis as follows:

$\mathbf{t}_i$ inserts a new color anticolor pair $c_{n+1}$ into the basis. This is done such that $c_i$ is connected to $\bar{c}_{n+1}$, and $c_{n+1}$ is connected to the original anticolor partner.

$\bar{\mathbf{t}}_i$ inserts a new color anticolor pair $c_{n+1}$ into the basis by invoking $\mathbf{t}_{\sigma^{-1}(i)}$, i.e., inserts a new color anticolor pair into the color line connected to the anticolor line it is applied to.

$\mathbf{s}_i$ inserts a color singlet.

Using these operators, we replace the usual DIRE leading color splitting kernels [28]

$$\begin{aligned} P_{i=q/\bar{q}, j=g, k}(z) &= C_{\text{F}} \left( \frac{2(1-z)}{(1-z)^2 + \kappa^2} - (1+z) \right) , \\ P_{i=g, j=g, k}(z) &= \frac{C_{\text{A}}}{2} \left( \frac{2(1-z)}{(1-z)^2 + \kappa^2} - 2 + z(1-z) \right) \end{aligned} \tag{9}$$

by color- and flow-specific kernels

$$\begin{aligned} P_{i=q, j=g, k}^{(9, c)}(z) &= \mathbf{T}_k \boldsymbol{\tau}_{ij}^c \frac{P_{i=q, j=g, k}(z)}{\mathbf{T}_{ij}^2} , & P_{i=\bar{q}, j=g, k}^{(\bar{9}, \bar{c})}(z) &= \mathbf{T}_k \boldsymbol{\tau}_{ij}^{\bar{c}} \frac{P_{i=\bar{q}, j=g, k}(z)}{\mathbf{T}_{ij}^2} , \\ P_{i=q/\bar{q}, j=g, k}^{(1, c)}(z) &= \mathbf{T}_k \boldsymbol{\tau}_{ij}^{s_c} \frac{P_{i=q/\bar{q}, j=g, k}(z)}{\mathbf{T}_{ij}^2} , \\ P_{i=g, j=g, k}^{(+, c)}(z) &= \mathbf{T}_k \boldsymbol{\tau}_{ij}^c \frac{P_{i=g, j=g, k}(z)}{\mathbf{T}_{ij}^2} , & P_{i=g, j=g, k}^{(-, \bar{c})}(z) &= \mathbf{T}_k \boldsymbol{\tau}_{ij}^{\bar{c}} \frac{P_{i=g, j=g, k}(z)}{\mathbf{T}_{ij}^2} , \end{aligned} \tag{10}$$

while the kernel

$$P_{i=g, j=q/\bar{q}, k}(z) = \frac{T_{\text{R}}}{2} (1 - 2z(1-z)) \tag{11}$$

remains unchanged except for updating the color basis to reflect the splitting of the gluons color and anticolor index onto the resulting quark antiquark pair. Each kernel thus fixes the inserted color $c \in 1, 2, 3$ and one of the flows, while the other flow is initially summed over to determine the branching rate, to then pick a specific flow for the further evolution as described below.

The QED splitting kernels for $q \to q\gamma$ and $\gamma \to q\bar{q}$ are adapted to update the color flow basis states of the respective states. While the former transfers the color line to the resulting quark, the latter creates a new color anticolor singlet dipole when splitting the photon into a pair of quark and antiquark.

Interference is taken into account by introducing two color vectors $\langle \mathcal{M}_n(\sigma', c_n)|$ and $|\mathcal{M}_n(\sigma, c_n)\rangle$, representing distinct color flows for a fixed set of color indices $c_n$. Every step in the evolution allows for a non-vanishing color flow pair to be chosen stochastically by evaluating the insertion of generators between $\langle \mathcal{M}_n(\sigma', c_n)|$ and $|\mathcal{M}_n(\sigma, c_n)\rangle$. The fixed state is kept for further evolution. At leading color, the color flow in both states is identical. Beyond leading color, all partons may be assigned as spectator, not just the leading-color connected ones. To tame the growth in complexity, we revert to the leading color structure in the shower and continue the shower evolution in a leading color fashion below a chosen cutoff in the shower evolution variable $t_{\mathrm{FC}}^{\mathrm{cut}}$.

The color- and flow-specific kernels in eq. 10 mean that the gluon emission probability in a fixed color shower step can, when normalizing by the "Born color" factor $|\mathcal{M}_n(c_n)|^2 = \sum_{\sigma,\sigma'} \langle \mathcal{M}_n(\sigma', c_n)|\mathcal{M}_n(\sigma, c_n)\rangle$, and inserting into a pre-determined color- and flow-structure, be written as

$$
\begin{aligned}
&\frac{1}{|\mathcal{M}_n(c_n)|^2} \sum_{ij,k\neq ij} \langle \mathcal{M}_n| \frac{-\mathbf{T}_k \mathbf{T}_{ij}}{\mathbf{T}_{ij}^2} |\mathcal{M}_n\rangle P_{i,j,k} \\
&= \frac{1}{|\mathcal{M}_n(c_n)|^2} \sum_{ij,k\neq ij} \sum_{\sigma,\sigma',c_n} \langle \mathcal{M}_n(\sigma', c_n)| \frac{-\mathbf{T}_k \mathbf{T}_{ij}}{\mathbf{T}_{ij}^2} |\mathcal{M}_n(\sigma, c_n)\rangle P_{i,j,k} \\
&= \frac{1}{|\mathcal{M}_n(c_n)|^2} \sum_{ij,k\neq ij} \sum_{\sigma,\sigma',c_n} \sum_c \langle \mathcal{M}_n(\sigma', c_n)| \frac{-\mathbf{T}_k (\boldsymbol{\tau}_{ij}^c + \boldsymbol{\tau}_{ij}^{\bar{c}} + \boldsymbol{\tau}_{ij}^{s_c})}{\mathbf{T}_{ij}^2} |\mathcal{M}_n(\sigma, c_n)\rangle P_{i,j,k} \\
&= \frac{1}{|\mathcal{M}_n(c_n)|^2} \Bigg( \sum_{ij\in q,k\neq ij} \sum_{\sigma,\sigma',c_n} \sum_c \langle \mathcal{M}_n(\sigma', c_n)| P_{i=q,j=g,k}^{(9,c)} + P_{i=q,j=g,k}^{(1,c)} |\mathcal{M}_n(\sigma, c_n)\rangle \\
&\quad + \sum_{ij\in\bar{q},k\neq ij} \sum_{\sigma,\sigma',c_n} \sum_c \langle \mathcal{M}_n(\sigma', c_n)| P_{i=\bar{q},j=g,k}^{(\bar{9},\bar{c})} + P_{i=\bar{q},j=g,k}^{(1,c)} |\mathcal{M}_n(\sigma, c_n)\rangle \\
&\quad + \sum_{ij\in g,k\neq ij} \sum_{\sigma,\sigma',c_n} \sum_c \langle \mathcal{M}_n(\sigma', c_n)| P_{i=g,j=g,k}^{(+,c)} + P_{i=g,j=g,k}^{(-,\bar{c})} |\mathcal{M}_n(\sigma, c_n)\rangle \Bigg)
\end{aligned}
\tag{12}
$$

In the first step, we decompose the state into the Monte Carlo sampled states employed in this fixed color shower algorithm, and the last step shows the decomposition of the gluon emission dipole term into the kernels employed in the fixed color shower, sorted by the flavour of the emitting parton.

The generation of each kernel's contribution, for a fixed radiator-recoiler pair, to the total color correlator and the resulting change in (no-)emission rates,

$$
\langle \mathcal{M}_n(\sigma', c_n)| \frac{-\mathbf{T}_k \boldsymbol{\tau}_{ij}^{\beta}}{\mathbf{T}_{ij}^2} |\mathcal{M}_n(\sigma, c_n)\rangle ,
\tag{13}
$$

is implemented by using a weighted parton shower algorithm [34–36]. More specifically, the usual leading color coefficients are used for an initial sampling, and an additional accept/reject step is added to correct the distribution to the desired fixed color result. The weights are calculated from the desired distribution $f$, the initially sampled leading color distribution $h$ and a conveniently chosen auxiliary overestimate $g$

$$
\begin{aligned}
f &= -\langle \mathcal{M}_n(\sigma', c_n)| \mathbf{T}_k \cdot \boldsymbol{\tau}_{ij}^{\beta} |\mathcal{M}_n(\sigma, c_n)\rangle \\
h &= \phantom{-}\langle \mathcal{M}_n(\sigma, c_n)| \mathbf{T}_{ij}^2 |\mathcal{M}_n(\sigma, c_n)\rangle \\
g &= \begin{cases} -h & \text{if } f/h < 0 \\ 2f & \text{if } |f/h| > 1 \\ h & \text{otherwise} \end{cases}
\end{aligned}
\tag{14}
$$

Splittings are accepted with probability $\frac{f}{g}$, and receive the accept weight $\frac{g}{h}$ or the reject weight $\frac{g}{h}\frac{h-f}{g-f}$. The former weight gives the desired fixed color distribution for this splitting, and the latter is used for rejected splittings, which are

then exponentiated appropriately. The implementation of color sampling exchanges the $ijk$ and $\sigma, \sigma'$ summation in eq. (12), thus fixing the color flow on which splitting of $i$, $j$ and $k$ can act. This makes it necessary to probabilistically sample a specific color flow after a splitting has occurred, such that this Monte-Carlo sum recovers eq. (12). A specific color flow is thus sampled according to

$$P = \frac{P_{\alpha\beta}}{\sum_\alpha P_{\alpha\beta}} \qquad \text{with} \qquad P_{\alpha\beta} = \left| \langle \mathcal{M}(\sigma', c_n) | \boldsymbol{\tau}_k^\alpha \cdot \boldsymbol{\tau}_{ij}^\beta | \mathcal{M}(\sigma, c_n) \rangle \right| . \tag{15}$$

As defined above, $\boldsymbol{\tau}$ represents the possible color insertions, both nonet and singlet, and the absolute value ensures a non-negative term. The appropriate sign and value are then restored by applying the following weight to accepted emissions:

$$\frac{g_{\text{col}}}{h_{\text{col}}} = \frac{\boldsymbol{\tau}_k^\alpha \cdot \boldsymbol{\tau}_{ij}^\beta}{|\boldsymbol{\tau}_k^\alpha \cdot \boldsymbol{\tau}_{ij}^\beta|} \frac{\sum_\alpha |\boldsymbol{\tau}_k^\alpha \cdot \boldsymbol{\tau}_{ij}^\beta|}{\sum_\alpha \boldsymbol{\tau}_k^\alpha \cdot \boldsymbol{\tau}_{ij}^\beta} . \tag{16}$$

In our implementation in DIRE , we choose to register multiple copies of the kernels described in eq. (10), each for a fixed choice of the new fundamental color index. This simplifies the construction of parton shower histories that we need for the kinematic matrix element corrections described in section II C. Note that the new splitting kernels compete in the veto algorithm, thus fixing the color flow in the amplitude and the new fundamental color index. However, the color flow in the conjugate amplitude is not fixed by the kernel. Instead, every fixed color kernel still allows for the probabilistic sampling of the conjugate flow. Still, the associated new fundamental color index must agree with that of the amplitude color flow, as otherwise, the product would vanish. Also, suppose a nonet/singlet emission kernel is chosen. In that case, the conjugate amplitude will only be able to contain a singlet/nonet flow if the new color index and the emitting one agree. The pair of color flows chosen in a successful splitting is then kept for further evolution.

It should be noted that, while this approach introduces subleading corrections in $1/N_{\text{C}}$, the emissions are still modelled by factorized dipole radiation. As mentioned in the introduction, if there are $N_{\text{C}}$ suppressed contributions due to interference effects between different dipole emission patterns, as is the case in $e^+e^- \to q\bar{q}q\bar{q}$ with identical flavors, these will not be reproduced by the shower. Kinematic matrix element corrections, as described in the following, can be used to include such effects. The same holds for the interference between states that can have both a QCD and a QED like emission history, which is also the case for $e^+e^- \to q\bar{q}q\bar{q}$.

## C. Kinematic Matrix Element correction to fixed color states

QCD and QED evolution are both implemented in the DIRE parton shower approach. However, since separate and competing splitting kernels govern them, the interference between full color QCD and QED does not enter by itself. To include the interference, we employ iterated matrix element corrections following the approach described in [29] and extend that method to produce corrections that respect the fixed color flow choices sampled in the shower, instead of mapping the matrix elements to color ordered states compatible with the shower's leading color picture. We summarize the present leading color matrix element corrections and then detail the changes necessary to apply matrix element corrections to the fixed color parton shower.

First, let us introduce the shorthand $f_n$ to denote a set of flavor identifiers, and define a phase-space point $\Phi_n = \Phi_n(p_n, c_n, f_n)$ as a fixed point in in momentum-, flavor- and color-space. With this, we redefine the notation for color-space vectors to also incorporate momentum- and flavor-dependence $\langle \mathcal{M}_n(\sigma, c_n) | \to \langle \mathcal{M}_n(\sigma, p_n, c_n, f_n) | = \langle \mathcal{M}_n(\sigma, \Phi_n) |$, such that the latter is indeed the full matrix element for a fixed color flow. We define the transition probability to a fixed phase space point *and* fixed flow as $|\mathcal{M}(\sigma, \Phi_n)|^2 = \langle \mathcal{M}_n(\sigma, \Phi_n) | \mathcal{M}_n(\sigma, \Phi_n) \rangle$. The remaining dependence on the color flow, and requiring an identical flow in bra and ket vector means that these transition probabilities are accurate only at the leading-color level. We will first review leading-color matrix-element corrections before discussing fixed-color results.

Parton shower emissions are governed by splitting kernels $P$ which can symbolically written as $P = P(\Phi_{n+1}/\Phi_n)$, assuming that the $(n+1)$-parton configuration $\Phi_{n+1}$ can be obtained from the $n$-parton configuration $\Phi_n$ by a parton shower branching. The fixed-order rate with which a parton shower generates the state $\Phi_{n+1}$ from underlying states $\Phi_n$ of fixed color flows is given by

$$|\mathcal{M}_{\text{PS}}(\sigma_{n+1}, \Phi_{n+1})|^2 = \sum_{\sigma, \Phi_n} \frac{8\pi\alpha(\mu_{\text{R}})P(\Phi_{n+1}/\Phi_n)}{Q^2(\Phi_n)} |\mathcal{M}(\sigma, \Phi_n)|^2 , \tag{17}$$

i.e., the rate is given by the branching probability from all underlying states from which the desired configuration is reachable. Here, $Q^2$ denotes the virtuality of the emitter-emission pair, and the coupling factor can represent the

strong coupling $\alpha_{\mathrm{s}}$ or the electromagnetic coupling $\alpha_{\mathrm{em}}$, depending on the nature of the splitting kernel. The sum over $\Phi_n$ only runs over states from which the shower could have generated the desired configuration $\Phi_{n+1}$, i.e. has to respect the ordering condition of the shower.

Symbolically, matrix element corrections implement the correction of these parton shower emission rates to the accurate leading-color matrix elements $|\mathcal{M}(\sigma_{n+1}, \Phi_{n+1})|^2$ by applying a correction $\mathcal{R}$:

$$\mathcal{R}(\sigma_{n+1}, \Phi_{n+1}) \sum_{\sigma, \Phi_n} \frac{8\pi\alpha(\mu_{\mathrm{R}})P(\Phi_{n+1}/\Phi_n)}{Q^2(\Phi_n)}|\mathcal{M}(\sigma, \Phi_n)|^2 \quad \text{with} \quad \mathcal{R}(\sigma_{n+1}, \Phi_{n+1}) = \frac{|\mathcal{M}(\sigma_{n+1}, \Phi_{n+1})|^2}{\sum\limits_{\sigma', \Phi_n'} \frac{8\pi\alpha(\mu_{\mathrm{R}})P(\Phi_{n+1}/\Phi_n')}{Q^2(\Phi_n')}|\mathcal{M}(\sigma', \Phi_n')|^2}. \tag{18}$$

where the sum over $\Phi_n'$ only covers ordered phase space configurations (and color flows $\sigma'$) produced by the shower, and $\mu_{\mathrm{R}}$ is a reference renormalization scale. Summing over all possible emissions in the first term of eq. (18) then ensures that the total parton shower rate reproduces the fixed order matrix element as desired. Note that the correction factor depends on the new phase space configuration and all possible underlying configurations $\Phi_n'$, i.e., its value is identical for every possible emission path.

For iterated ordered emissions, the same idea is applied. In contrast to a single emission, there are then many more possible parton shower emission sequences that could lead to a given state. In order to construct the correction factor, all possible parton shower histories leading to the given state from any underlying initial Born configuration need to be constructed. For the second parton shower emission, the correction factor is constructed as follows:

$$\mathcal{R}(\sigma_{n+2}, \Phi_{n+2}) = \frac{|\mathcal{M}(\sigma_{n+2}, \Phi_{n+2})|^2}{\sum\limits_{\sigma_{n+1}', \Phi_{n+1}'} \frac{8\pi\alpha(\mu_{\mathrm{R}})P(\Phi_{n+2}/\Phi_{n+1}')}{Q^2(\Phi_{n+1}')} \mathcal{R}(\sigma_{n+1}', \Phi_{n+1}') \sum\limits_{\sigma'', \Phi_n''} \frac{8\pi\alpha(\mu_{\mathrm{R}})P(\Phi_{n+1}'/\Phi_n'')}{Q^2(\Phi_n'')}|\mathcal{M}(\sigma'', \Phi_n'')|^2} \tag{19}$$

The rate for a certain state $\Phi_{n+2}$ after two emissions is then given by

$$\mathcal{R}(\sigma_{n+2}, \Phi_{n+2}) \sum_{\sigma_{n+1}, \Phi_{n+1}} \frac{8\pi\alpha(\mu_{\mathrm{R}})P(\Phi_{n+2}/\Phi_{n+1})}{Q^2(\Phi_{n+1})} \mathcal{R}(\sigma_{n+1}, \Phi_{n+1}) \sum_{\sigma, \Phi_n} \frac{8\pi\alpha(\mu_{\mathrm{R}})P(\Phi_{n+1}/\Phi_n)}{Q^2(\Phi_{n+1}/\Phi_n)}|\mathcal{M}(\sigma, \Phi_n)|^2. \tag{20}$$

Summing over all possible emissions again ensures that the emission rate for that phase space point corresponds to the desired (leading-color) matrix element.[1]

Matrix element corrections may also be used to improve the transition rate of the fixed-color shower proposed in the last section,

$$P_{\mathrm{FC}}(\Phi_{n+1}/\Phi_n) = \frac{1}{|\mathcal{M}_n(c_n)|^2} \sum_{\sigma, \sigma'} \langle \mathcal{M}_n(\sigma', c_n)| \frac{-\mathbf{T}_k \mathbf{T}_{ij}}{\mathbf{T}_{ij}^2} |\mathcal{M}_n(\sigma, c_n)\rangle P_{i,j,k} , \tag{21}$$

where the color structure $c_n$ is extracted from $\Phi_n$, as is the set of radiators $ij$ and the recoilers $k$ (cf. eq. (12)). Improving this transition rate with matrix element corrections allows us to incorporate fixed-color effects introduced by the interference of different dipole emission patterns or between amplitudes of different coupling structure – neither of which can, in general, be produced by using a step-by-step shower approach alone. To achieve matrix-element corrections for the fixed-color showers presented above, it is necessary to avoid matrix elements that artificially rely on color flows, and instead use matrix elements that only rely on phase-space points (i.e. fixed points in momentum-, flavor- and color (index) space) $\Phi_n$. We thus define $\langle \mathcal{M}_n(\Phi_n)| = \sum_\sigma \langle \mathcal{M}_n(\sigma, \Phi_n)|$ and $|\mathcal{M}(\Phi_n)|^2 = \langle \mathcal{M}_n(\Phi_n)|\mathcal{M}_n(\Phi_n)\rangle$. Replacing the flow-specific transition probabilities into the matrix element correction factors $\mathcal{R}$ (see eqs. 18 and 19) leads to the color-index specific matrix element corrections, e.g.

$$\mathcal{R}(\Phi_{n+1}) = \frac{|\mathcal{M}(\Phi_{n+1})|^2}{\sum\limits_{\Phi_n'} \frac{8\pi\alpha(\mu_{\mathrm{R}})P_{\mathrm{FC}}(\Phi_{n+1}/\Phi_n')}{Q^2(\Phi_n')}|\mathcal{M}(\Phi_n')|^2} \tag{22}$$

$$\mathcal{R}(\Phi_{n+2}) = \frac{|\mathcal{M}(\Phi_{n+2})|^2}{\sum\limits_{\Phi_{n+1}'} \frac{8\pi\alpha(\mu_{\mathrm{R}})P_{\mathrm{FC}}(\Phi_{n+2}/\Phi_{n+1}')}{Q^2(\Phi_{n+1}')} \mathcal{R}(\Phi_{n+1}') \sum\limits_{\Phi_n''} \frac{8\pi\alpha(\mu_{\mathrm{R}})P_{\mathrm{FC}}(\Phi_{n+1}'/\Phi_n'')}{Q^2(\Phi_n'')}|\mathcal{M}(\Phi_n'')|^2} , \tag{23}$$

---

[1] In DIRE , these matrix element corrections are implemented using a weighted shower algorithm similar to the one mentioned in eq. (14). With appropriate choices of $f$, $g$ and $h$, the accept and reject probabilities and weights are constructed as above.

where $P_{\text{FC}}$ denotes splitting kernels of the fixed-color shower. These correction factors only rely on the phase space points themselves, and no longer specific color flows. To construct the required color-index specific kinematic matrix elements $\langle \mathcal{M}_n(\Phi_n)|$, we use the fact that the partial amplitudes in the fundamental and the color-flow decomposition are closely related. Since the fundamental color indices (with values 1, 2 or 3) are sampled in the shower, we can pass fundamental color indices for the evaluation of the amplitude instead of passing a specific flow. The evaluation of the amplitudes then works as follows:

- Do not include the color matrix associated with the color-ordered amplitudes, as the color sum is performed in an MC fashion by the parton shower.

- For all-quark final states: check that the leading color flow induced by the ordering of the partial amplitudes are compatible with the fundamental indices produced by the shower, and only allow such amplitudes to contribute.

- For gluon amplitudes: Since there are no singlet gluons in the color-ordered amplitudes, we need to project external gluons onto a nonet and singlet component. For gluons with distinct fundamental indices, we take the same approach as for quarks. For gluons that carry the same fundamental index, we need to apply the projection

$$P_{jj'}^{ii'} \equiv \delta_{j'}^i \delta_j^{i'} - \frac{1}{N_{\text{C}}} \delta_j^i \delta_{j'}^{i'} \, . \tag{24}$$

In addition to the regular contribution based on compatibility of color ordering and fundamental indices, we get contributions that consider the adjacent partons. For the cross term between the first and the second, the fundamental indices of the gluon need to agree, as well as the adjacent ones according to the color ordering. For the second term squared, the fundamental indices of the gluon must be identical, as must be the fundamental indices of adjacent partons.

We combine the fixed color correction and the kinematic matrix element correction by applying two successive accept/reject steps. These come on top of the usual leading color veto algorithm accept/reject step, giving three accept/reject steps in total. However, the additional steps are only done if the previous step was accepted. In other words, the corrective fixed color weight is only applied to accepted leading color shower emissions, and the kinematic matrix element correction is only constructed if a parton shower emission was not rejected in the fixed color correction. For applying the kinematic matrix element correction on top of the fixed color shower, the construction of $\mathcal{R}$ furthermore needs to take into account the color configuration sampled by the shower, leading to more complex parton shower histories as compared to a leading color parton shower history.[2] Figure 10 in appendix A schematically shows the iteration of multiple accept/reject steps for fixed color and kinematic matrix element corrections.

We use MADGRAPH5 to generate matrix elements, and adapt the automatically generated C++ code[3] to our needs. We limit the application of kinematic matrix element corrections to $e^+e^- \to jj$ states with one additional photon or gluon or states with four quarks in the final state. For the $q\bar{q}g$ state, the amplitude will be unmodified for distinct fundamental and anti-fundamental gluons, with two such states sampled by the shower (assuming one color is fixed by the initial process). For all-identical fundamental indices, we get a factor $1 - 2/N_{\text{C}} + 1/N_{\text{C}}^2$ with just one such configuration sampled. For a singlet gluon with identical fundamental color and anticolor indices that are different from the $q\bar{q}$ color index, there will be a factor $1/N_{\text{C}}^2$ with two such configurations sampled by the shower. Together with the factor $T_{\text{R}} = 1/2$, this gives the expected factor $C_{\text{F}} = 4/3$ as required. As the correction is based on fundamental indices, and the explicit cancellation of the chosen flow combinations in the shower needs to be retained, the MEC factors $\mathcal{R}$ are constructed to include the color weights constructed in 14, but not the additional sign-restoring weight factor in 16. The implementation of a more generic matrix element plugin with multi-gluon states would be desirable but is beyond the scope of this study. For the study of QCD/QED interference effects in $q\bar{q}q\bar{q}$ states, only configurations leading to this state need to be implemented.

Figure 2 shows example parton shower histories for a given state with four final-state quarks, also including QED histories. Due to singlet gluons in the color flow basis, there are now quark anti-quark pairs that can be clustered into both gluons and photons.

---

[2] The current shower sampled fixed color weight going into $f$, $g$, and $h$ can be neglected on the other hand, since all probabilities and weights are constructed as ratios of these.

[3] We would like to thank Valentin Hirschi for extending and developing PYTHIA-tailored C++ output methods in MADGRAPH5 .

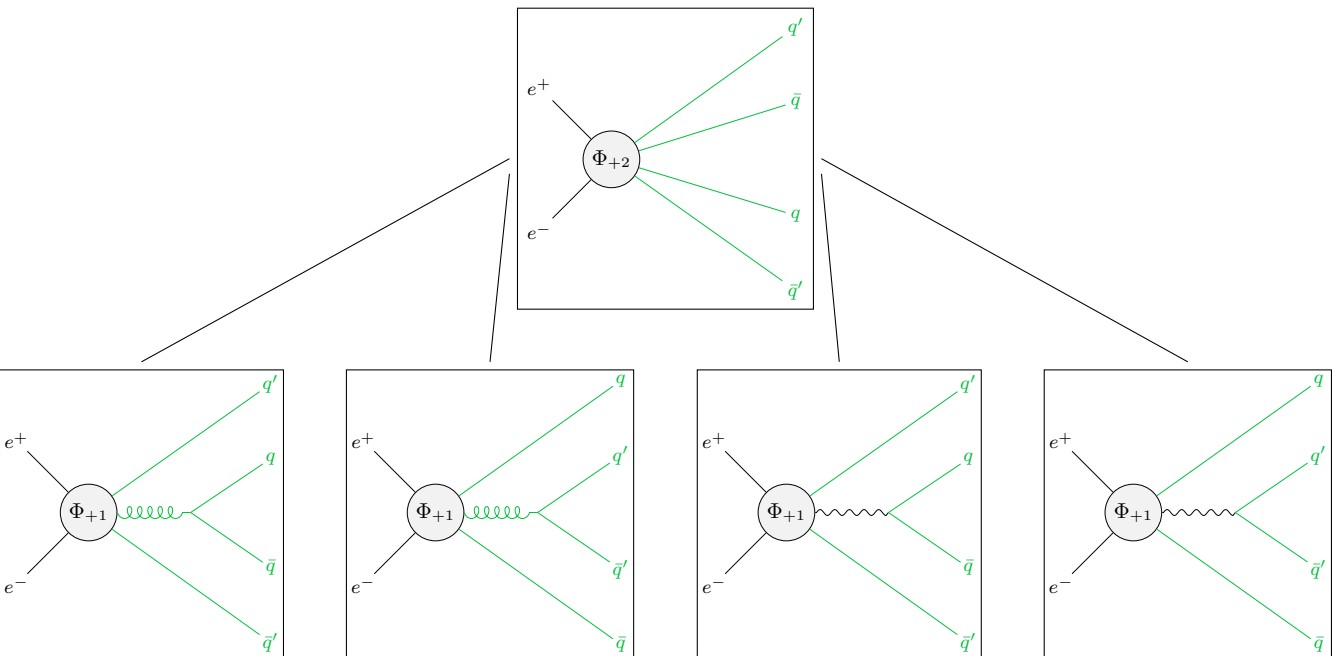

FIG. 2: Histories for two quarks and to anti-quarks in the final state in fixed color QCD and QED evolution, where all carry either green or anti-green fundamental indices: Each flavour pair can either be clustered into a gluon (left two) or a photon (right two). Only one or the other assignment would be suitable in a leading color picture, depending on whether the colors match (photon) or don't match (gluon). The exact flow is only sampled after all clustering steps are done.

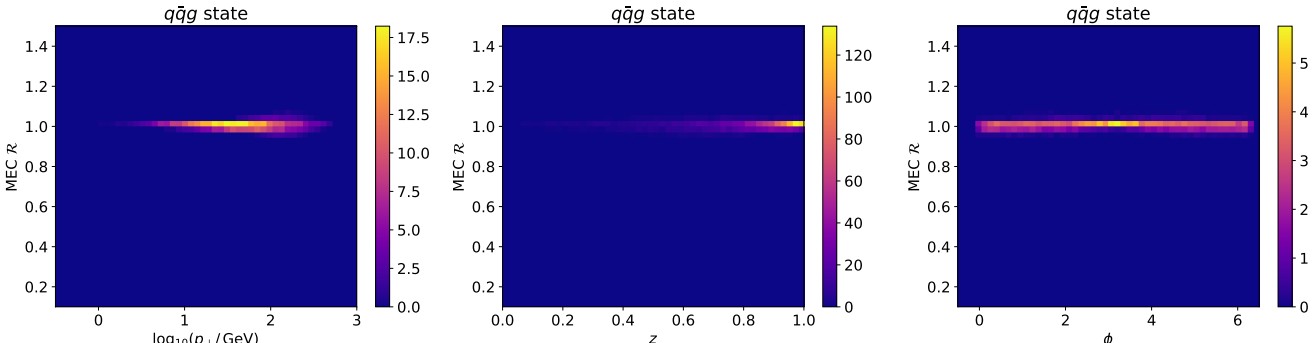

FIG. 3: Consistency validation of parton shower and matrix element description of $q\bar{q}g$ state, plotted against the shower transverse momentum, the energy fraction of the gluon and the azimuthal angle of the gluon emission. The color indicates the normalized sampling rate of the respective points. In this case, all phase space points are well approximated by the parton shower, such that the MEC factor $\mathcal{R}$ is always close to one.

## III. RESULTS

This section presents results on the parton level for the $e^+e^- \to q\bar{q}$ process with one or two further emissions. We focus on the $qq\bar{q}\bar{q}$ final state for two emissions by not allowing for a second photon or gluon emission from three-parton states. All plots are shown for a center of mass energy of 1 TeV.[4]

To check the consistency of the implemented fixed color shower evolution and the corresponding fixed color matrix element corrections, we plot the matrix element correction factor $\mathcal{R}$ for the states sampled by the shower against different kinematic variables. For the $q \to qg$ splitting, we get the distribution shown in fig. 3. The emission pattern

---

[4] The effects of a fixed color shower on this state as compared to a leading color shower correspond to $C_F$ vs $C_A/2$, so if the leading color shower uses $C_F = 4/3$, no difference is expected. Thus, we will show fixed-color results throughout.

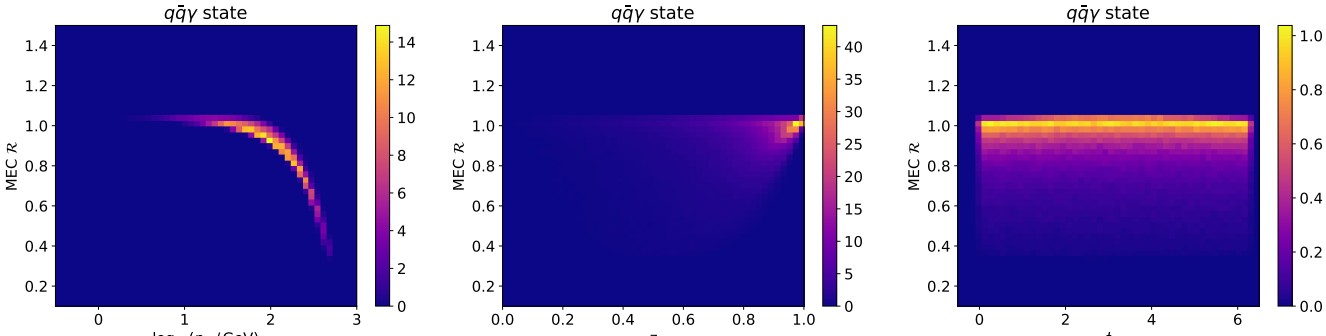

FIG. 4: Consistency validation of parton shower and matrix element description of $q\bar{q}\gamma$ state. The MEC factor $\mathcal{R}$ converges to one for low transverse momenta as required, but differs for larger transverse momenta, compensating for an enhanced photon emission rate by the shower.

of a first gluon is well modelled by the shower, such that the matrix element correction (MEC) factor is close to one for all sampled phase space points. A minimal requirement would be to see a convergence in the collinear region, i.e., for low transverse momenta.

For the emission of one photon, we see a similar situation, see fig. 4. However, due to the slight variation of the functional form for the QED emission as mentioned in eq. (3), the shower gives an enhanced photon emission rate for large transverse momenta as compared to gluon emissions. This enhancement is compensated by the application of kinematic matrix element corrections, such that the MEC factor $\mathcal{R}$ is below one for large transverse momenta. However, as required for a proper collinear shower approximation, the correction still converges to one in the collinear limit.

For the parton shower generation of two quarks and two anti-quark states, the corresponding kinematic matrix element corrections are shown in fig. 5. The matrix element correction for this state is now based on the shower rate for two successive branchings, leading to a wider spread in the applied corrective factors. Nevertheless, the correction is still centered around one, especially for low transverse momenta in the second splitting. Furthermore, the correction factor shows a clear structure in the azimuthal angle. This is expected because the shower emissions are based on averaged splitting functions, while the kinematic MEC contains azimuthal correlations due to the intermediate vectors.

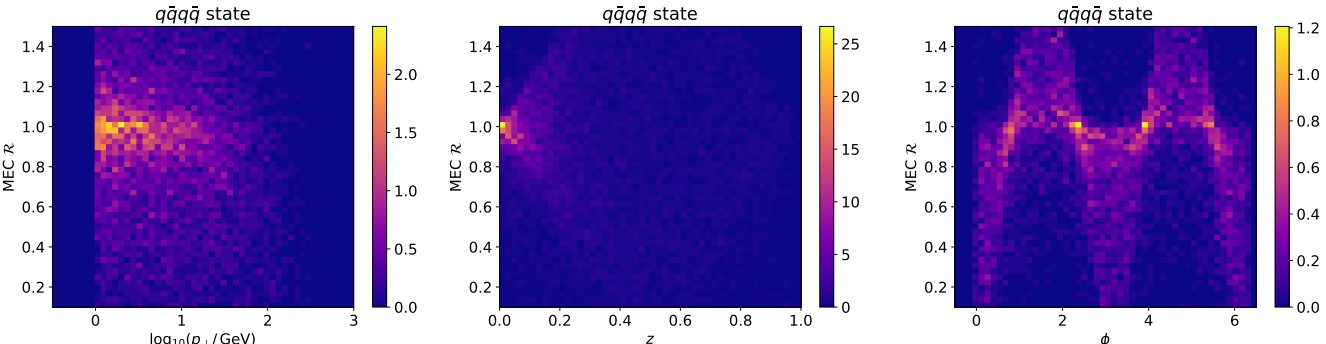

FIG. 5: Consistency validation of $qq\bar{q}\bar{q}$ state. For small transverse momenta, the correction is centered around one. An azimuthal dependence of the correction can be seen, correcting the averaged emission pattern of the shower.

Overall, this comparison shows that the fixed-color and QED showers behave as expected, furnishing a sensible approximation in the soft/collinear limits. However, the poorer description of $q\bar{q}q\bar{q}$ states indicates that the effect of MECs on observables may be appreciable.

Before looking at the effect of QCD/QED interference in matrix-element-corrected $qq\bar{q}\bar{q}$ states, we would like to examine the distinct ingredients of the simulation. All following plots were generated using RIVET [39]. Figure 6 shows the effect of including photon emission in the (uncorrected) parton shower evolution, as compared to just including QCD splittings. Keeping in mind that we enhance large transverse momentum QED emissions as described in eq. (3), we see an effect of up to five percent in the Durham jet clustering scale of one additional parton, i.e., a gluon or a photon emitted from the quark anti-quark dipole in the final state. Initial state radiation is not considered in this context. In the right plot, we see that the invariant mass of the pair of energetically softer quark and anti-quark in

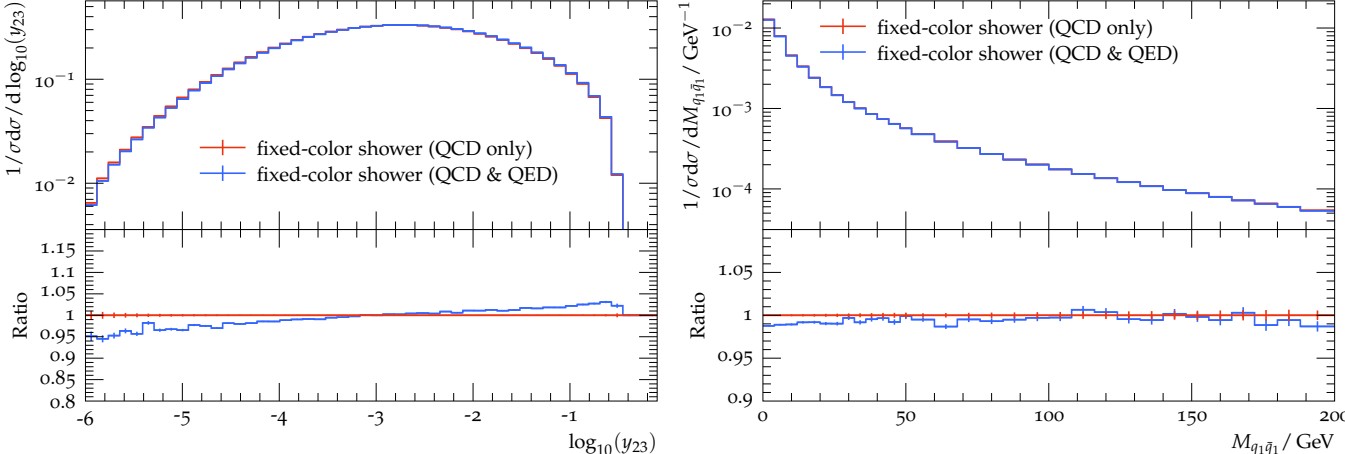

FIG. 6: Comparison of pure QCD shower evolution and shower evolution with photon emissions enabled. The left plot shows the Durham jet clustering scale for the emission of one gluon or photon. The inclusion of photon emissions shifts the spectrum towards higher scales, as expected. The right plot shows the invariant mass of the pair of less energetic quark and anti-quark for the $qq\bar{q}\bar{q}$ state. The effect of the QED emission alone is very small, just leading to a light depletion of very small configurations.

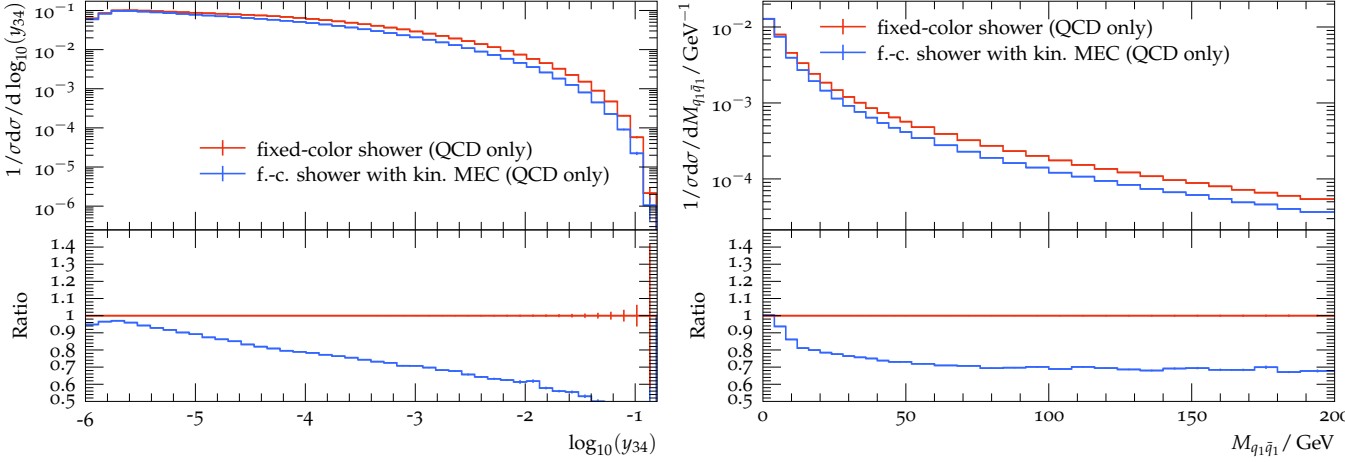

FIG. 7: Effect of kinematic matrix element corrections on the $qq\bar{q}\bar{q}$ state for QCD evolution and correction only. The left plot shows the $4 \to 3$ Durham jet clustering scale, while the right plot shows the invariant mass of the pair of energetically softer quark and anti-quark. We see a large effect of up to 50% for very high jet masses.

$qq\bar{q}\bar{q}$ states is only mildly affected by including photons in the parton shower evolution. In most cases, this observable would represent the invariant mass of the quark anti-quark from the gluon or photon splitting, making it sensitive to potential interference effects.

The effect of enabling matrix element corrections on the $q\bar{q}q\bar{q}$ state for QCD only is shown in fig. 7. Surprisingly, we see that the shower significantly over-samples the state as compared to the matrix element prediction. Thus, including the matrix element correction has an effect of up to 50% on the resulting distribution. This shows that the effect of including kinematic matrix element corrections for this specific configuration is much larger than the effect of photon emissions and branchings in the shower, which is expected due to the smallness of the electromagnetic coupling.

As shown in section I, the $q\bar{q}q\bar{q}$ state is interesting for finding interference effects between QCD and QED for the combined evolution with interference-sensitive kinematic matrix element corrections, which would not be present in the $q\bar{q}q'\bar{q}'$ state. The interference between same-color $q\bar{q}$ pair from gluon and photon splittings is expected to vanish due to their octet and singlet nature. However, a QCD/QED interference effect at leading color is expected for same-flavour $q\bar{q}q\bar{q}$ pairs. We thus compare the $q\bar{q}q\bar{q}$ and $q\bar{q}q'\bar{q}'$ states employing the complete machinery of mixed QED shower and QCD+QED matrix element corrections to look for possible interference signatures. Figure 8 shows

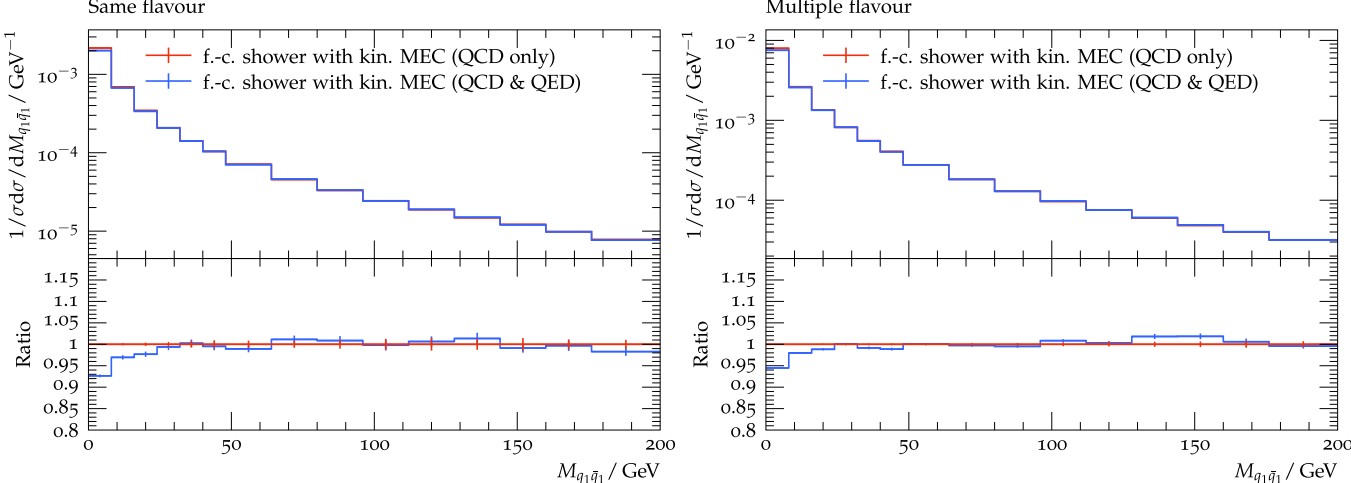

FIG. 8: Both plots show the invariant mass of the pair of less energetic quark and antiquark at 1 TeV. The left plot shows $e^+e^- \to q\bar{q}q\bar{q}$, while the right shows $e^+e^- \to q\bar{q}q'\bar{q}'$. QCD/QED interference effects are only expected on the left side. However, no qualitative difference is observed between left and right, indicating that the QCD/QED interference effect in this observable is negligible.

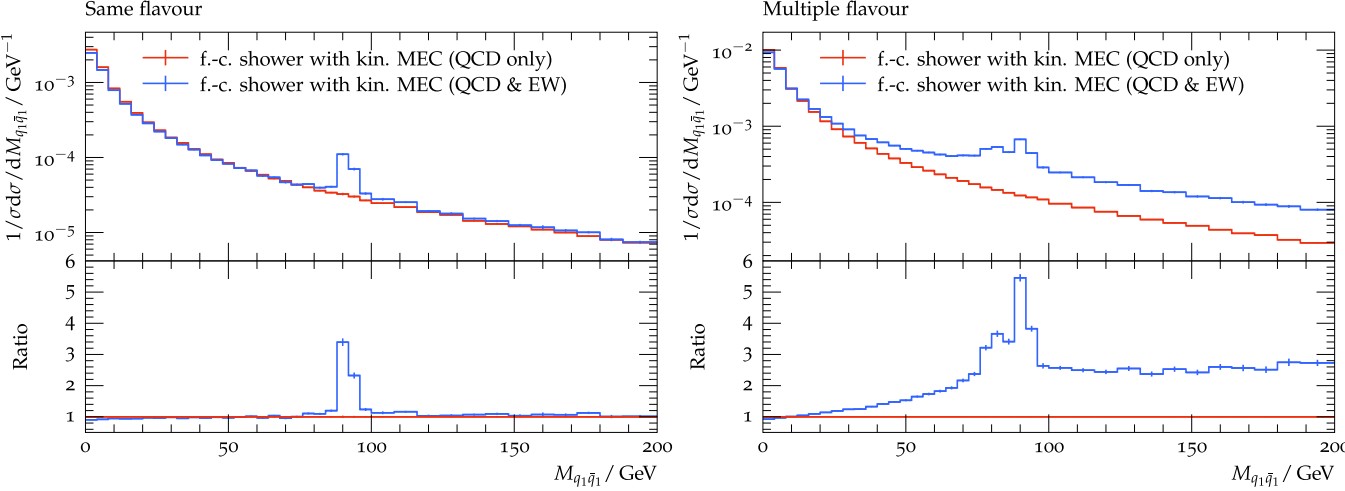

FIG. 9: Same distributions as shown in fig. 8, but including the effect of $W^\pm$, $Z$ and $H$ bosons. For the considered CM energy of 1 TeV, the effects are very large, showing resonance-like structures for both same- and different-flavour configurations, and an overall enhanced distribution for the different-flavour case.

the resulting distributions for MEC corrections, including QCD and QED states. The total rate between same-flavour and different-flavour pairs differs, but the qualitative shape compared between QCD and QCD+QED matrix element corrections does not show a large effect. Most interestingly, when comparing the same-flavour and different-flavour states, the ratio plots indicate that the difference between QCD and QCD+QED corrections in each is very similar. This seems symptomatic for the large set of observables we have investigated during this study and indicates that the QCD/QED interference, which could have been shown in the same flavour case, appears to be negligible.

In order to further put the observed effects into context, we show the same corresponding distributions in fig. 9, where we take into account $W^\pm$, $Z$ and $H$ bosons in the employed matrix element correction instead of just including QED effects. In the resonance regions, we see effects of up to 350%, while the distributions at high momenta are also affected by a factor of up to 2.5. This clearly shows that the modelling of electroweak effects plays a significant role beyond subleading color, QED or QCD/QED interference effects for high collision energies.

## IV. CONCLUSIONS AND OUTLOOK

Precision event generators will likely become ever more important for high-energy physics. Quite often, precision calculations of SM backgrounds focus on sophisticated QCD corrections. This is also true for parton showering, which has seen many QCD-focussed improvements in recent years.

The goal of this article was to instead assess interferences between processes with different coupling structures within the parton shower. Such a comparison will either lead to a confirmation of a QCD-focussed correction strategy or highlight shortcomings thereof. To perform a consistent comparison, and given that interference effects may contribute to different color configurations with different amount, we have developed a complex parton shower framework that incorporates fixed-color QCD evolution, QED evolution including all possible soft-photon enhancements, corrections to full tree-level branching rates for individual color configurations through matrix element corrections, and finally incorporating QCD/QED interference through iterated matrix-element corrections.

We have investigated the different effects using the first non-trivial process with sufficient structure, $e^+e^- \rightarrow q\bar{q}q'\bar{q}'$. Overall, the result supports a QCD-focussed parton-shower strategy to parton-shower improvements, given that QCD/QED interference effects are surprisingly small. The effect of kinematic matrix element corrections was nevertheless enormous in some cases due to probing (the square of) diagrams containing new electroweak resonances.

The developments in this article can be helpful in various aspects of parton showers in the future. The implementation of a fixed-color shower within PYTHIA opens the door to extending matching (and merging) methods beyond leading color, since it will make fixed-color hard-scattering events viable starting points for shower evolution. Since the fixed-color treatment is also intertwined with QED emissions, a consistent QED/QCD matching appears feasible. Such developments could be aided by new insights into color flows [40].

Various phenomenological studies could help refine our understanding of (the smallness of) QCD/QED interference effects. In particular, it would be interesting to study interference effects in the presence of initial-state QED radiation. The kinematic structure of the latter is quite different from the final-state radiation topologies we have focussed on in this article.

## V. ACKNOWLEDGMENTS

We acknowledge funding from the European Union's Horizon 2020 research and innovation programme as part of the Marie Skłodowska-Curie Innovative Training Network MCnetITN3 (grant agreement no. 722104). This note is supported by funding from the Swedish Research Council, contract numbers 2016-05996 and 2020-04303.

### Appendix A: Implementation details

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
