# Peer review of "Coloring mixed QCD/QED evolution"

_SciPost Physics_

## Round 1 · Referee Report · Anonymous · 2021-11-27

Report

In this paper the author discuss a parton shower algorithm that combines the QED and QCD effect in the DIRE framework. More specifically they are interested in the interference contributions of the non-trivial QCD colour structures and the similar QED amplitudes as it is nicely illustrated in Eq.~(1). In order to do this they have to deal with the colour evolution in the parton shower algorithm. This is very complicated task since the colour basis is not orthogonal and a dimensionality of the problem is enormous. It scales with the number of external legs as $\sim (n!)^2$.
The QED part of their shower implementation looks reasonable in a simple shower framework where the spins of the particles are averaged.

The major part of this project is to deal with colour in the parton shower. This procedure is based on the algorithm that is defined in of Ref.[14]. That is a published paper in Phys.Rev.D. and in principle this would make my job easier. I should simply accept this paper, but I cannot recommend it without major revision since I have several concerns.

\subsection*{QED contributions}

In Sec.~II.A there is review of the QED part of the Dire parton shower. In the introduction the author pointed out that the interference terms between the QED and QCD radiations has to be understood and treated very carefully. In Section II.A there are only two type of QED vertices are discussed, the photon emission one ($q\to q+\gamma$), $P_{\mathrm{q},\gamma,k}$, and the photon splitting vertex ($\gamma\to q+\bar{q}$), $P_{\gamma, \mathrm{q}, k}$. These vertices are leading contributions in $\alpha_{em}$ and they can describe the $\alpha_{em}^2$ terms in Eq~(1) after two steps of the shower.

The parton showers algorithms evolves diagonal states in the momentum and flavour space. In a first order shower, which is discussed here, in every step of the shower procedure we can generate only one extra parton, thus the interference diagrams of Eq.~(1) (the last two graphs) are completely unreachable by the first order shower. With first order shower we cannot generate a gluon on the bra side and a photon on the ket side of the matrix element square in the first step as it is suggested in the last two terms of Eq.~(1). That requires higher order contributions in the parton shower algorithm. Namely a splitting kernel that is proportional to $\alpha_s\alpha_{em}$,
%
\begin{equation*}
P^{(2)}\sim \alpha_s\alpha_{em} \langle \bm{V}^\dagger(q\to q+(g\to q'+\bar{q}'))\cdot \bm{V}(q\to q+(\gamma\to q'+\bar{q}'))\rangle,
\end{equation*}
%
where $\bm{V}$ is the Feynman graph that describes the given partonic process. The $\langle\cdots\rangle$ is the spin averaging.
This is the territory of the NLO shower algorithms with real and virtual QCD and QED corrections to splitting kernels.

{\bf These higher order contributions are clearly not in this shower algorithm, thus it cannot deal with the QED/QCD interference terms. Could you please clarify this in the text?}

\subsection*{The colour evolution, Sec. II.B}

The main idea here and in Ref.[14] is to go beyond the usual leading colour approximation in a systematic way and sum up some subleading colour effect ($1/N_c^2$) correctly. As far as I know that requires amplitude level evolution in the colour space.

The authors in this paper use the colour flow basis and in this case any $m$-parton amplitude can be written as
%
\begin{equation*}
\big|{\cal M}_m\big\rangle = \sum_{\sigma} |\sigma\rangle M_m(\sigma)\;\;,
\end{equation*}
%
where $M_m(\sigma)$ are the colour subamplitude in the colour flow basis.

If we want to do colour evolution on this basis, for the shower evolution equation we have something like\footnote {I hope this equation is clear enough, I tried to stick to the notation of the paper. For the sake of simplicity the $z$ and azimuth integrals and the momentum mappings are included in $P_{i,j,k}(\mu^2)$ kernels and I ignored the $g\to q+\bar{q}$ vertex as well as the QED vertices.}
%
\begin{equation*}
\begin{split}
&\mathbb{S}_{\mu^2_{\rm cut}, \mu^2_{m}}\big[|{\cal M}_m\rangle \langle{\cal M}_m|\big] =
\Delta(\mu^2_{\rm cut}, \mu^2_{m})|{\cal M}_m\rangle\langle{\cal M}_m|\Delta^\dagger(\mu^2_{\rm cut}, \mu^2_{m})
\\
&+\int_{\mu^2_{\rm cut}}^{\mu^2_{m}}\frac{d\mu^2}{\mu^2}\sum_{i,k=1\atop i\neq k}^m\mathbb{S}_{\mu^2_{\rm cut}, \mu^2}\Big[
\bm{T}_i\Delta(\mu^2, \mu^2_{m})|{\cal M}_m\rangle \frac{P_{i,m+1,k}(\mu^2)}{\bm{T}_i^2}\langle{\cal M}_m|\Delta^\dagger(\mu^2, \mu^2_{m})\bm{T}_k^\dagger
\Big].
\end{split}
\end{equation*}
%
Here $\mathbb{S}_{\mu^2_{1}, \mu^2_{2}}[\cdots]$ represents shower evolution between two scales. In the first term we have evolution between the scale of the $M_m$ state and the final cutoff scale without any real emission. While in the second term we have no emission from $\mu^2_m$ to $\mu^2$ and then we have a real emission. After this we can continue the shower with more emission by $\mathbb{S}_{\mu^2_{\rm cut}, \mu^2}[\cdots]$. Surely, all the amplitudes $|{\cal M}_m\rangle$ has to be expanded on the colour basis.

{\bf It would be nice to have a precise definition of the shower evolution operator/equation in the paper, especially there are extra complications from the QED part.}

\paragraph{Question I: What represents the partonic state?}

If we want to do colour evolution it has to be the density matrix in the colour space,
%
\begin{equation*}
\big|{\cal M}_m\big\rangle \big\langle{\cal M}_m\big| = \sum_{\sigma,\sigma'} |\sigma\rangle\langle\sigma'| \, M_m(\sigma)M_m(\sigma')^*\;\;,
\end{equation*}
%
where $|\sigma\rangle$ is the colour basis vector as it is given in Eq.~(7). The paper suggests that the object that is evolved by the shower evolution is the matrix element square,
%
\begin{equation*}
\big\langle{\cal M}_m \big|{\cal M}_m\big\rangle\;\;,
\end{equation*}
%
but in this case a consistent colour evolution is impossible. Without the precise definition of the shower evolution equation it is hard see how it can happen.

\paragraph{Question II: What is the real splitting operator?}

If we take seriously the colour evolution the real splitting operator should be something like
%
\begin{equation*}
P_{i,m+1,k}(\mu^2) \frac{\bm{T}_i|\sigma\rangle \langle\sigma'| \bm{T}_k}{\bm{T}_i^2}
\end{equation*}
%
when it is act on a $|\sigma\rangle \langle\sigma'|$ color state. One can expand the $\bm{T}_i$ operator on the colour flow basis and express this operator in terms of $\bm{t}_c\otimes \bm{t}_{c'}$, $\bm{t}_c\otimes \bar{\bm{t}}_{c'}$, $\bm{t}_c\otimes \bm{s}$,.... In Eq.(10) the splitting operator depends on the colour operators only linearly. This is not clear at all. Since we have bra and ket states and they have to be evolved independently in the colour space.

\paragraph{Question III: What is the Sudakov operator?}

The Sudakov operator is a timed ordered exponential of the splitting kernel of the virtual and unresolvable radiation, thus qualitatively we should have something like this:
%
\begin{equation*}
\Delta(\mu^2, \mu^2_{m}) = \mathbb{T}\exp\left(
-\frac{1}{2}\int_{\mu^2}^{\mu^2_{m}}\frac{d\bar\mu^2}{\bar\mu^2} \sum_{i,k=1\atop i\neq k}^m \frac{\bm{T}_i\cdot\bm{T}_k}{\bm{T}_i^2} P_{i,m+1,k}(\bar\mu^2)
\right)\;\;.
\end{equation*}
%
The Sudakov operator is a nontrivial exponentiated operator in the colour space and it sums up virtual and unresolvable radiation at all order level. Since the colour basis is not orthogonal and dimension of the colour space is enormous, this exponentiation is rather hopeless. Thus, it important to have a precise definition of the Sudakov operator. Furthermore, the QED part of the Sudakov operator is not even mentioned in this manuscript.

The unitarity is an important property of the parton shower algorithms and that relies on the consistent definition of the real emission operator and the Sudakov operator.

As far as I understood their Sudakov operator is something like
%
\begin{equation*}
\begin{split}
&\Delta(\mu^2, \mu^2_{m})|\sigma\rangle \langle\sigma'| \Delta^\dagger(\mu^2, \mu^2_{m})
\\
&= |\sigma\rangle \exp\left(
-\int_{\mu^2}^{\mu^2_{m}}\frac{d\bar\mu^2}{\bar\mu^2} \sum_{i,k=1\atop i\neq k}^m
\frac{\langle\sigma'|\bm{T}_i\cdot\bm{T}_k|\sigma\rangle }{\langle\sigma'|\bm{T}_i^2|\sigma\rangle} P_{i,m+1,k}(\bar\mu^2)
\right)\langle\sigma'|
\;\;.
\end{split}
\end{equation*}
%
I think it is obvious why it is wrong. One can achieve unitarity by this, but most important and primary role of the Sudakov operator is to sums up virtual and unresolvable emissions at all order level. Expanding this Sudakov in $\alpha_s$ the generated term don't correspond to any Feynman graphs in the strongly ordered soft and collinear limits. This Sudakov is not even the first approximation of the proper Sudakov operator. It is kinda acceptable only in the strict leading colour limit, beyond that it completely ad-hoc and messes up completely the $1/N_c^2$ subleading colour effects.

Attachment

  • validity: high
  • significance: good
  • originality: ok
  • clarity: low
  • formatting: -
  • grammar: perfect

Author:  Stefan Prestel  on 2021-12-06  [id 2012]

(in reply to Report 1 on 2021-11-27)

#Response to 1st Report

We would like to acknowledge the work of the Referee in producing detailed comments. Unfortunately, it appears that some crucial aspects of the manuscript might have gone unnoticed when preparing the review. Thus, we would like comment on the points raised by the referee, hoping that this clarifies the goals and methods of the project.

**Referee comment: QED Contributions**
It is not necessary to resort to NLO parton showers to embed interference. Any iterated shower may produce the final state with a known rate, which can be re-weighted to the correct rate, including the interference between QCD and QED. The shower acts as phase-space generator, which produces momentum configurations on which any squared (sum of) matrix elements (including interferences) may be evaluated. We use iterative matrix element corrections, as described in II.C, to achieve this. We refer the Referee to this crucial aspect of the manuscript -- which is not mentioned in the report -- to resolve the misunderstanding.

Once we have also received comments from the second referee, we aim at including an additional sentence in the manuscript to again stress the importance of matrix element corrections for our approach.

**Referee comment: The colour evolution, Sec. II.B**
The Referee's comments from this point on seem to assume that we are aiming for amplitude-level evolution. While amplitude-level evolution might be a tool to consistently sum up $1/N_c^2$ effects, we are rather building upon the factorization of the real-emission pattern at the cross-section level. The color-flow basis is used as a way to sample the fundamental colour indices of the produced states, which characterize the partonic state that is to be corrected through matrix-element corrections.

Once we have also received comments from the second referee, we aim at including an additional sentence to stress that no amplitude-level evolution is attempted.

**Referee Question I: What represents the partonic state?**
The referee seems to assume here that we aim to perform amplitude-level evolution, and in fact that amplitude-level evolution is the only possible tool to embed subleading-$N_c$ corrections within parton showers. The vast amount of literature concluding differently (see e.g. the small selection \cite{Gustafson:1992uh,Friberg:1996xc,Platzer:2012np,Platzer:2018pmd,Hamilton:2020rcu}), in particular when focusing on improved real-emission descriptions in showers, puts this assessment in doubt.

To address the question in the section title: The partonic state is represented by a set of four-momenta, flavors and fundamental colour indices for the final state partons.

**Referee Question II: What is the real splitting operator?**

The splitting kernels employed in the fixed-color parton shower are explained in detail in sec. II.B, in particular in eqs. 8-11. Further details are given in [14], and in the flowchart in the appendix. The correct color-correlator prefactor for each splitting is instated by the procedure explained on page 5. As a purely implementational method, the color correlator is calculated by performing the colour insertion independently on auxiliary bra and ket states, with non-vanishing combinations being accepted. This calculational aspect does, however, not mean that any amplitude-level evolution algorithm is employed.

Once we have also received comments from the second referee, we aim at including an formula that summarizes the splitting probability after the procedures described in sec II.B and II.C.

**Referee Question III: What is the Sudakov operator?**
The unitarity property of the parton shower relies on a consistent sampling algorithm, which is based on the colour-corrected real-emission pattern given by the splitting kernels. The reweighting procedure for fixed-colour states and matrix element corrections described in the paper allows for the correction of these emission patterns, while preserving the unitarity of the shower. We do not claim to do consistently exponentiated amplitude level evolution in this manuscript. Instead, the main aim of the manuscript was to assess if an $1/N_c$-improved description of real-emission patterns was phenomenologically meaningful without considering also QED effects, as argued in detail in the introduction.

The referee also comments on a Sudakov operator of their own definition. This definition assumes the application of a type of amplitude-level evolution. Since this is at odds with the methods and goals of the manuscript, we refrain from commenting on this issue.

##Bibliography

\bibitem{Gustafson:1992uh}
G.~Gustafson,
\newblock Nucl. Phys. B {\bf 392}, 251 (1993).

\bibitem{Friberg:1996xc}
C.~Friberg, G.~Gustafson, and J.~Hakkinen,
\newblock Nucl. Phys. B {\bf 490}, 289 (1997), hep-ph/9604347.

\bibitem{Platzer:2012np}
S.~Platzer and M.~Sjodahl,
\newblock JHEP {\bf 07}, 042 (2012), 1201.0260.

\bibitem{Platzer:2018pmd}
S.~Pl\"atzer, M.~Sjodahl, and J.~Thor\'en,
\newblock JHEP {\bf 11}, 009 (2018), 1808.00332.

\bibitem{Hamilton:2020rcu}
K.~Hamilton, R.~Medves, G.~P. Salam, L.~Scyboz, and G.~Soyez,
\newblock (2020), 2011.10054.

Attachment:

Scipost_1st_Report_Response.pdf

---

## Round 1 · Referee Report · Anonymous · 2021-12-6

Strengths

1. An important step in the development of parton shower algorithms aiming at incorporating interferences of processes with different coupling structures, which is important for precision Monte Carlo simulations of processes, within the Standard Model and Beyond, studied at high-energy physics colliders, such as the LHC or the planned ILC/CLIC/FCC.
2. The first fixed-color parton shower implementation with the iterative color-specific kinematic matrix element corrections in the PYTHIA Monte Carlo generator.
3. Provides important calculations and results on the QCD/QED interference effects in the process of electron-positron annihilation into hadrons.

Weaknesses

1. Some minor shortcomings in the figures and in the text, as listed in the "Requested changes" section.

Report

The presented manuscript addresses the issue of QCD/QED interference effects in parton shower simulations of high-energy particle collisions. The authors concentrate on the process of quark-antiquark pair production in electron-positron annihilation with final-state emission of a gluon, a photon, or an additional quark-antiquark pair. This process is considered mainly as a test case, in order to reveal the important aspects of the problem and its proposed solution, and as a starting point for further developments.

In the Introduction, by performing a simple analysis based on the Feynman diagrams for the process of double quark-antiquark pair production in electron-positron annihilation, the authors argue that the QCD/QED interference contributions can be of the size of the subleading QCD corrections, and even accidental cancellation between these two effects might occur. This motivates them to incorporate the QCD/QED interferences into a QCD-focused parton shower algorithm. In order to do this, they need to go beyond a typical leading-color approximation of a QCD parton shower algorithm and to use fixed-color parton shower evolution as well as to include color-specific kinematic matrix element corrections (MECs). Details of the implementation together with technicalities of Monte Carlo simulations are described in Section II and Appendix A. All the necessary features are implemented with the help of the dedicated fixed-color parton shower algorithm DIRE (developed by S. Prestel et al.), used as a plugin to the PYTHIA Monte Carlo event generator.

Numerical results are presented in Section III. First, the authors perform the consistency check of the implemented fixed-color parton shower evolution for the considered parton splittings by presenting the matrix element correction factor for different kinematical variables. These results show that the pertinent fixed-color and QED showers provide adequate approximations for the exact amtrix elements, particularly in the soft/collinear limits, although in the case of four-quark final states, MECs can be large in some kinematical regions. In the following plots, the authors demonstrate that including QED splittings affects rather mildly the parton shower evolution (effects up to 5%), while the QCD MECs effects can be large (up to 50%). Then, in Fig. 8 the authors show the size of the QCD/QED interference effects by comparing the invariant mass distributions of the less energetic quark-antiquark pair for the same-flavor and multiple-flavor cases. These contributions turn out to be negligible, particularly when compared with the electroweak effects (coming from the W, Z and H boson contributions), shown in Fig. 9.

In my opinion, this paper constitutes an important step in the development of parton shower algorithms aiming at incorporating interferences of processes with different coupling structures, which is important for precision Monte Carlo simulations of processes, within the Standard Model and Beyond, studied at high-energy physics colliders, such as the LHC or the planned ILC/CLIC/FCC. The authors present the first fixed-color parton shower implementation with the iterative color-specific kinematic matrix element corrections in the PYTHIA Monte Carlo generator. The considered QCD/QED interferences turned out to be numerically negligible in the studied case, but the authors made significant progress in the development of the parton shower algorithm that accounts for this type of effect. Therefore, I recommend this manuscript for publication in the SciPost Physics journal, however after introducing the (minor) corrections as listed below.

Requested changes

1. In Eq. (1), after the 1st term in the 2nd line, the dot (before +) should be removed.
2. Plots in Figs. 1-5 are a bit too small - in the printed version of the paper they are barely legible.
3. In Figs. 3-5, the authors do not specify the meaning of the 3rd axis (corresponding to colors). If it corresponds to the number of generated events, it would be better - in my opinion - to normalize the corresponding histograms to 1.
4. In Figs. 3-5, left plots: in the horizontal axis, p_T should be divided by the physical unit (e.g., p_T/GeV), as the argument of the log-function should be dimensionless.
5. In Figs. 6-9, the physical unit (GeV, I suppose) for the invariant mass in the horizontal axis is missing.
6. In Fig. 8, there is a spike in the ratio distribution at the invariant mass value of 140 which probably results from statistical fluctuations - I would recommend generating a higher statistics sample in order to clarify this.
7. In Fig. 9, the ratio distribution in the RHS plot seems to be cut at the top - the scale in the vertical axis is too small.
8. On page 7, in the 3rd line of the 2nd paragraph the authors write: "the amplitude will be modified by a factor 1". Does this mean that it will be unmodified, or something is missing there?
9. Two lines below the above, the phrase "with different but identical quark color indices" sounds like a contradiction.
10. On page 6, in footnote 2, the authors introduce the abbreviation FC without explanation of its meaning, similarly for the abbreviations LC and MEC on page 7.
11. Section II.A, in the 1st line of the 2nd paragraph: "evolution variables" -> "evolution variable".
12. Appendix A: in the bottom left box: "hisories" -> "histories".

---

## Editorial Decision

resubmitted